# Relationship between Fall History and Self-Perceived Motor Fitness in Community-Dwelling People: A Cross-Sectional Study

**DOI:** 10.3390/jcm9113649

**Published:** 2020-11-13

**Authors:** Katsushi Yokoi, Nobuyuki Miyai, Miyoko Utsumi, Sonomi Hattori, Shigeki Kurasawa, Hiroko Hayakawa, Yuji Uematsu, Mikio Arita

**Affiliations:** 1Graduate School of Comprehensive Rehabilitation, Osaka Prefecture University, Osaka 583-8555, Japan; 2School of Health and Nursing Science, Wakayama Medical University, Wakayama 641-0011, Japan; miyain@wakayama-med.ac.jp (N.M.); sonomi@wakayama-med.ac.jp (S.H.); hayakawa@wakayama-med.ac.jp (H.H.); yujiue@wakayama-med.ac.jp (Y.U.); 3Wakayama Faculty Nursing, Tokyo Healthcare University, Wakayama 640-8269, Japan; m-utsumi@thcu.ac.jp; 4Faculty of Health Sciences, Kansai University of Welfare Sciences, Kashiwara 582-0026, Japan; kurasawa@tamateyama.ac.jp; 5Sumiya Rehabilitation Hospital, Wakayama 640-8344, Japan; arita@sumiya.or.jp

**Keywords:** fall history, history of multiple falls, motor fitness

## Abstract

History of falling is an important fall risk factor. If a relationship between fall history and self-perceived motor fitness could be established, then treating it as a correctable risk of re-fall due to falls may be possible. We conducted a cross-sectional study of the relationship between fall history and self-perceived motor fitness in daily life among 670 community-dwelling people (mean age 62.0 ± 9.6 years, 277 men and 393 women) who had participated in health examinations. They completed a self-administered questionnaire that asked about their history of single or multiple falls and included a 14-item motor fitness scale. The responses were analyzed using multivariate logistic regression analysis. The results showed that in both younger and older (<65 years) subjects, a history of single or multiple falls was associated with a negative response to “being able to put on socks, pants or a skirt while standing without support”. For subjects ≥65 years, an association was also observed with “shortness of breath when climbing stairs”. Self-perceived motor fitness related to fall history can easily be noticed by an individual and may help them become aware of fall-related factors earlier in everyday life.

## 1. Introduction

Falls among older people is an important public health problem [1]. People who fall may require hospitalization; falls can negatively affect the lives of older people and subsequently impact their activities of daily living (ADL) and quality of life (QOL) [2,3]. As lifespans increase, fall becomes increasingly important, including addressing correctable fall risk factors.

Fall risk factors include intrinsic risks associated with an individual’s traits and extrinsic risks, such as the environment [4]. Most falls are due to a combination of age, illness-related health status and interactions between individuals and their social and physical environments [1]. Among them, history of falls is an important risk factor for its reoccurrence [5,6]. Older people who have experienced a fall tend to refrain from activities due to the fear of it reoccurring, which reduces their ability to perform ADL and diminishes the QOL [7]. Fear of falling often arises after a fall and is a fall risk. A relationship between fear of falling and a history of falling has previously been reported [8]. Moreover, a history of falls has been shown to be associated with poor balance, as assessed by the Berg balance scale (BBS) and the timed up and go test (TUG) [9,10]. The gait characteristics of older people with a history of falls include reduced minimum foot clearance, a critical event in the gait cycle as the foot travels with a maximum horizontal velocity around this instant [11]. Furthermore, falls are associated not only with physical aspects but also with loneliness, social isolation and social frailty related to living alone [12]. A history of multiple previous falls has been linked to falling again [13]. Therefore, fall history occupies an important position in the clinical algorithm for falls [14] and constitutes an essential part of screening for falls. Because of the possibility of being able to detect fall risk early, assessing motor fitness while carrying out everyday activities that can be observed by people themselves and related to a history of falls can be significant. A previous study has developed a self-perceived motor fitness scale (MFS) that focuses on activities closely related to daily life, including 14 items (climbing stairs, walking, carrying luggage, changing clothes, etc.). MFS may also be used as a substitute for physical performance when assessing the physical function of community-dwelling older people [15].

Previously, fall history has been reported to be associated with placing an alternate foot on a stool and standing with one foot in front; however, a relationship between fall history and specific motor fitness has not been demonstrated [8]. Moreover, while attention has been paid to fall history among older people, it has not been adequately examined in people less than 65 years old. There are only a handful of studies related to falls targeting middle-aged and older people [16].

This study investigated the relationship between fall history and motor fitness among community-dwelling people aged between 40 and 75 years.

If a relationship between fall history and motor fitness can be identified, then older people may be able to easily identify fall-related factors in their lives.

## 2. Methods

### 2.1. Participants

The subjects were 674 people who had participated in the 2015 Wakayama Health Promotion Study and had undergone an arteriosclerosis health examination conducted in Katsuragi, Wakayama Prefecture, in the southern part of the Kinki region in Japan. In the Wakayama Health Promotion Study, doctors, nurses, public health nurses and university and medical researchers surveyed lifestyle diseases using data from community health examinations performed by the town government for residents aged 40–75 years old [17]. The subjects did not include hospital inpatients or people living in institutions. They were required to be able to walk into the health examination venue on their own. All those who had undergone the health examination agreed to participate in the study. The final analysis set comprised of 670 people, excluding 4 people who did not completely fill out the questionnaire on daily activities. The questionnaires were distributed in advance and checked for missing items on the day of the health examination. If missing items were found, the subjects were asked to complete them on that day.

This study was approved by the Wakayama Medical University ethics committee (approval number: 92, approval date: 8 July 2011), and both oral and written explanations of the study were given to the participants before obtaining their consent.

### 2.2. Basic Attributes

A self-administered questionnaire distributed in advance asked about sex, age, education history (elementary school, middle school, high school, junior college, university), diseases (hypertension, diabetes, cerebrovascular disease, bone or joint disease, cataracts, heart disease, anemia), urinary incontinence, types of medication and alcohol consumption. Body mass index (BMI) was calculated from the height and weight measured at the health examination. As malnutrition is associated with falling [18], the albumin level was surveyed from a blood test performed at the health examination.

### 2.3. Fall Survey

A self-administered questionnaire asked the respondents whether they had a fall in the past year [5], and those with a history of falls were asked how many times they had fallen. A fall was defined as “unintentionally coming to the ground or some lower-level other than as a consequence of sustaining a violent blow, loss of consciousness, sudden onset of paralysis as in stroke or an epileptic seizure” [19].

### 2.4. Self-Perceived Motor Fitness Scale (MFS)

The MFS Japanese version is a questionnaire that surveys the motor fitness of older people in daily life [15]. The predictive power of physical function assessed by the MFS is related to the assessment of physical performance in community-dwelling older people [15]. It comprises 14 items: 6 on mobility, 4 on strength and 4 on balance. Each item is answered by either “yes” (1 point) or “no” (0 points), with total scores ranging from 0 to 14 points. The higher the score, the higher the degree of motor function. The test-retest validity of the MFS has been confirmed, and it has also been shown to have criterion-related validity [15]. In a study among community-dwelling older people, lower MFS scores were associated with a significantly higher incidence of needing long-term nursing care 4 years later, and those with low MFS scores (men ≤ 11 points, women ≤ 9 points) had a 3.04 times greater risk of newly requiring nursing care [20].

### 2.5. Grip Strength

Grip strength was measured alternately in the left and right hands, in the standing position with the elbow extended, on the day of the health examination, using a Smedley hand dynamometer (KK5401, Takei Scientific Instruments Co., Niigata, Japan). The higher value was used to represent grip strength [21].

### 2.6. Social Frailty

While not clearly defined, social frailty refers to a state of increased vulnerability concerning participation in social activities and social interactions [12]. For the assessments, we used the Japanese version of the social frailty screening index [22], based on the concept of social frailty described by Bunt et al. [23]. Self-administered questionnaires were distributed in advance and checked for missing items on the day of the health examination. The questionnaire comprised 4 items that assessed: whether the person had financial difficulties, representing general resources; whether the person was living alone, representing social resources; whether the person took part in events or activities in neighboring communities, representing social participation; and how the person interacted with neighbors, representing basic social activities. A state of social frailty is when 2 or more of these 4 items apply, pre-social frailty when 1 item applies, and socially robust when 0 items apply.

### 2.7. Fear of Falling

We selected the question, “are you anxious about falling at home?” from the 25-question geriatric locomotive function scale (GLFS-25) created by the Japanese Orthopedic Association. This was answered with a 5-point scale: not anxious, slightly anxious, moderately anxious, very anxious and extremely anxious [24]. The questions were asked in advance in a self-administered questionnaire.

### 2.8. Statistical Analysis

First, the characteristics of the subjects were compared according to the presence or absence of fall history. Due to limited studies related to falls in middle-aged and older people [16], the subjects were classified as <65 years old or ≥65 years old. Descriptive statistics were summarized for a history of single or multiple falls as the dependent variables and basic attributes, MFS, grip strength, social frailty and fear of falling as the independent variables. Comparisons were performed using the Mann–Whitney U test and chi-squared test. Next, multivariate logistic analysis (forced entry method) was used to examine how the presence or absence of a history of single or multiple falls are related to the independent variables. A history of multiple falls was analyzed for subjects ≥65 years of age. Of the independent variables, the 14 MFS items were binarized as “yes” or “no,” with “yes” (ability to perform the activity in the question) as the reference. Social frailty was classified as socially robust, pre-frailty or frailty [22]. Fear of falling was binarized as “not anxious” and “anxious”. Three exploratory models were created for the multivariate logistic regression analysis. Model 1 interactively adjusted the 14 MFS items. Model 2 was Model 1 plus an adjustment for the potential confounding factors of sex, educational history, alcohol consumption, BMI, types of medication, urinary incontinence and diseases. Model 3 was Model 2 plus an adjustment for grip strength, which is associated with risk of falling; and albumin levels, which is associated with malnutrition, social frailty and fear of falling. The variance inflation factors (VIF) of all the independent variables were in the 1.02 to 2.87 range, and no multicollinearity was observed. The software SPSS Ver. 26 (IBM, Tokyo, Japan) was used to perform the statistical analyses, and the level of significance was set at 0.05.

## 3. Results

The final analysis set was 670 people (mean age 62.0 ± 9.6 years, 277 men, 393 women). In the past year, 103 (15.4%) patients had experienced a fall. This was 38 of 330 (13.0%) in the <65 years group and 65 of 340 (23.6%) in the ≥65 years group. The number of people with multiple falls was 16 (5.0%) in the <65 years group and 31 (9.1%) in the ≥65 years group. No difference was noted between men and women in all groups (Table 1).

Table 2 and Table 3 show the relationship between the history of falling and MFS. A history of falling was significantly associated with the inability to “put on socks, pants or a skirt while standing up without support” in both <65 years and ≥65 years groups. In the <65 years group, the adjusted odds ratios (OR) were 4.18 (95% confidence interval CI: 1.11–15.80) for Model 1, 5.96 (95% CI: 1.27–28.00) for Model 2 and 5.60 (95% CI: 1.03–30.55) for Model 3. In the ≥65 years group, the adjusted OR was 2.44 (95% CI: 1.18–5.04) for Model 1, 3.49 (95% CI: 1.55–7.89) for Model 2 and 3.90 (95% CI: 1.62–9.36) for Model 3. A significant association of history of falling with the inability to “go upstairs without becoming short of breath” was observed in the ≥65 years group. The adjusted OR was 2.27 (95% CI: 1.21–4.32) for Model 1, 2.51 (95% CI: 1.21–5.20) for Model 2 and 2.36 (95% CI: 1.08–4.32) for Model 3. In the ≥65 years group, A significant association of history of falling with the inability to “walk fast to catch up with someone walking” was observed only in Model 1.

A significant association of history of falls with the inability to “put on socks, pants or a skirt while standing without support” was observed among people ≥65 years old with a history of multiple falls in all models (Table 4).

## 4. Discussion

This was the first cross-sectional study investigating the relationship between the history of falling and motor fitness in daily life among community-dwelling people. One advantage of motor fitness is that it can be observed in daily life activities. To date, measuring physical function using equipment (timed up and go test [25], functional reach test [26], four square step test [27], etc.) was necessary for assessments related to falls. However, motor fitness does not require measurements with special equipment as it exists in daily life. In this study, 23.6% of older people had a history of falling in the past year, similar to previous studies in Japan [28] and can be regarded as a representative value. There is no comparable data on people <65 years old, making this study the first to provide it. A history of falling carries great weightage among the fall risk factors. Finding a relationship between fall history and motor fitness using activities such as changing clothes and climbing stairs could help people realize that they are at fall-related factors amid everyday life. In addition, the risk of falls may be detected at an early stage.

An association with fall history and “shortness of breath when climbing stairs” was observed for subjects ≥65 years old. Even if fall risk factors such as BMI, types of medication, disease and social frailty are adjusted, it was shown that climbing stairs should be an area of concern among older people. Stairs are an environment where the risk of falling is high [29]. However, almost no objective characteristics that can prevent falls have been identified [30]. A high step rate when descending stairs has been associated with falling at a later date [31]. However, little research has been conducted on behavioral interventions to reduce the risk when using the stairs [32]. As an environment, there is a strong relationship between movement characteristics when using the stairs and falling. Therefore, it is important to focus on the act of going up and down the stairs. Moreover, people with chronic obstructive pulmonary disease are known to have an increased risk of having balance disturbances and falling [33,34]. The present study did not include data on the respiratory organs, and thus, we cannot make any conclusions about the relationship between stair climbing and shortness of breath. However, we did demonstrate a relationship between a history of falling and shortness of breath when climbing stairs. This is a subjective symptom that appears in daily life and could be helpful because it is an easy method to use to become aware of fall-related factors.

Fall history is associated with changing clothes wherein the individual is required to stand on one leg. After the age of 60 years, the duration for which people are able to stand on one leg with eyes open decreases sharply: a duration of five seconds or less is considered to indicate a high risk of falling [35]. A cohort study that followed up 500 community-dwelling people for one year reported that short one-leg standing time was characteristic of people who experienced falls [36]. Furthermore, regarding the relationship between fall history and balance, an association with the BBS “standing with one foot in front” has been observed. An individual is required to stand on one leg while wearing socks or pants, but this differs from the normal one-leg standing in that it requires manipulating clothing and moving from the center of gravity in multiple directions. Although the time factor in one-leg standing is important, changing clothes is an observable event associated with the history of falling, and it can be checked by the individual. Physical activities and motor fitness are related to cognitive processes in older people [37], and the reduction in physical activity level and functional fitness was equal in both men and women due to the aging process [38]. The Otago exercise program, which has a high level of evidence as a fall prevention intervention, includes a one-leg standing exercise [39] without the need for tool operation. Various exercises performed in the one-leg standing position may be important. Furthermore, targeted interventions have been conducted on people with a fall history [40], and it may be necessary to consider interventions with actions similar to changing clothes. Practicing moving from the center of gravity while standing on one leg may help prevent falls. A survey on risky behaviors in daily life and risk of falls found that risky behaviors such as bending down and rushing to do things increased the risk of falls [41]. Associations with the “put on socks, pants or a skirt while standing without support” were observed in all models, for both people younger than 65 years and those older than 65 years. This can be considered the most responsive item, even when considering social frailty, which affects disease and falling. However, the large 95% confidence interval for the <65 years group indicates that this result should be interpreted carefully. An association between this item and a history of multiple falls was observed in people aged ≥65 years, indicating that this item deserves attention as a fall-related factor.

Focusing on the relationship between falls and movement, as about 50% of falls occur while walking [42], numerous studies have investigated the characteristics of falls and walking. Many have found that a walking speed of 1.0 m/s is a cutoff value for fall risk [43]. A study on walking speed and falls found that compared to a normal walking speed group (1.0 < 1.3 m/s), the risk of falling was higher in slow (<0.6 m/s) and fast (≥1.3 m/s) groups [44]. Older people who have experienced falls have shorter strides, larger fluctuations in cadence and walk slower than older people who have never fallen, which suggests that falling is linked not only to reduced walking speed but also to unstable walking [45]. In this study, a history of falling was associated with the inability to walk fast to catch up with someone. Catching up with someone requires walking slightly faster, which indicates a connection with walking speed. Instead of simply measuring the walking speed, this activity assesses a form of walking that is closely related to daily activities. However, no association with “walking fast to catch up with someone” was observed in people <65 years of age, whereas in people older than that, an association was observed only in Model 1. Walking is likely to be affected by factors other than a history of falling.

### 4.1. Clinical Significance

A history of falling is a strong risk factor for its reoccurrence. This study demonstrated that fall history is related to specific movements. Until now, specific motor fitness has not been shown to be among the intrinsic factors of fall risk. The actions that we found related to fall history are subjective symptoms and observable events, making it easier to become aware of fall-related factors. These findings may help detect the risk of falls at an early stage.

### 4.2. Limitations

This study has several limitations. First, confirming falls from recollection largely depends on the memory of older people, which may have created recall bias [46]. Although the recall method may underestimate the actual state of falls [47], its reliability has been largely confirmed as a survey method for community-dwelling older people [9]. Furthermore, fall history and all other measurements were performed simultaneously. The same method was used in previous studies [10]. In this study, all the measurement items, except for grip strength, were self-reported. Significantly, some of the motor fitness tasks reflected in daily life activities required clarification. Due to the estimation of motor fitness by a self-report, it is important that results from physical activity tests such as timed up and go or short physical performance battery be obtained [40]. Further studies are required in this regard. Second, there may have been other potential confounding factors as there was a lack of data on depression and cognitive functions related to falls. Additionally, the type of medication used and grip strength were adjusted in the analysis; however, indicators for lower limb muscle strength, visual impairment, static and dynamic balance were not used. Finally, although results can be generalized to relatively healthy community-dwelling residents who could come to the health center, they may not apply to residents of care facilities with different fall risk profiles compared to the general population. Furthermore, the age range was 40 to 75 years and did not include older people aged 76 years or older. Owing to the cross-sectional nature of the survey, it was impossible to demonstrate a causal relationship between fall history and motor fitness. In the future, a cohort study should be conducted on a larger age range to determine whether there are causal relationships between the history of single or multiple falls and daily activities.

## 5. Conclusions

Among community-dwelling people, a history of falling was associated with a negative response to “being able to put on socks, pants or a skirt while standing without support” and “being able to climb stairs without becoming short of breath”. To enable the early detection of fall risk, these motor fitness tasks reflected in daily life activities require further study to confirm their utility as fall-related factors.

## Figures and Tables

**Table 1 jcm-09-03649-t001:** Participant’s characteristics ^1^.

Self-Perceived Motor Fitness	Overall					<65 Years Old					≥65 Years Old				
	*n* = 670					*n* = 330					*n* = 340				
	No Fall History		Fall History		*p* Value	No Fall History		Fall History		*p* Value	No Fall History		Fall History		*p* Value
	*n* = 567		*n* = 103			*n* = 292		*n* = 38			*n* = 275		*n* = 65		
Age (years)	61.5	(9.7)	64.6	(8.5)	<0.01										
Sex															
Male	234	(41.3)	43	(41.7)	0.93	112	(38.4)	20	(52.6)	0.09	122	(44.4)	23	(35.4)	0.19
Female	333	(58.7)	60	(58.3)		180	(61.6)	18	(47.4)		153	(55.6)	42	(64.6)	
Education															
Elementary	1	(0.2)	0	0.0	0.06	0	0.0	0	(0.0)	0.03	1	(0.4)	0	0.0	0.20
Middle	56	(9.9)	8	(7.8)		12	(4.1)	2	(5.3)		44	(16.0)	6	(9.2)	
High	274	(48.3)	51	(49.5)		116	(39.7)	11	(28.9)		158	(57.5)	40	(61.5)	
Technical school	53	(9.3)	4	(3.9)		35	(12.0)	2	(5.3)		18	(6.5)	2	(3.1)	
Jr. college	77	(13.6)	21	(20.4)		63	(21.6)	13	(34.2)		14	(5.1)	8	(12.3)	
University	106	(18.7)	18	(17.5)		66	(22.6)	9	(23.7)		40	(14.5)	9	(13.8)	
Other	0	0.0	1	(1.0)		0	0.0	1	(2.6)		0	0.0	0	0.0	
Employment	344	(60.7)	66	(64.1)	0.51	223	(76.4)	31	(81.6)	0.47	121	(44.0)	35	(53.8)	0.15
Alcohol consumption	272	(48.0)	49	(47.6)	0.94	145	(49.7)	21	(55.3)	0.52	127	(46.2)	28	(43.1)	0.65
Medication (type)	2.1	(2.6)	3.1	(3.2)	<0.01	1.3	(2.1)	1.4	(2.0)	0.71	2.8	(2.8)	4.0	(3.4)	<0.01
Body mass index	22.5	(3.2)	22.9	(3.3)	0.24	22.3	(3.4)	22.5	(3.3)	0.72	22.6	(3.1)	23.1	(3.4)	0.29
Albumin (g/dL)	4.3	(0.2)	4.3	(0.2)	<0.01	4.4	(0.2)	4.3	(0.2)	0.12	4.3	(0.2)	4.3	(0.2)	0.04
Diseases															
Hypertension	174	(30.8)	44	(43.1)	0.01	47	(16.2)	7	(18.4)	0.73	127	(46.2)	37	(57.8)	0.09
Diabetes	39	(6.9)	10	(9.8)	0.31	10	(3.5)	1	(2.6)	1.00	29	(10.6)	9	(14.1)	0.43
Cerebrovascular disease	8	(1.4)	6	(5.8)	0.01	3	(1.0)	2	(5.3)	0.10	5	(1.8)	4	(6.2)	0.07
Heart disease	29	(5.1)	11	(10.7)	0.03	8	(2.7)	0	(0.0)	0.60	21	(7.6)	11	(16.9)	0.02
Bone-joint disease	31	(5.5)	15	(14.6)	0.01	15	(5.1)	1	(2.6)	1.00	16	(5.8)	14	(21.5)	<0.01
Cataracts	35	(6.2)	14	(13.6)	0.01	9	(3.1)	2	(5.3)	0.37	26	(9.5)	12	(18.5)	0.04
Anemia	66	(11.7)	9	(8.7)	0.39	53	(18.2)	3	(7.9)	0.11	13	(4.7)	6	(9.2)	0.22
Urinary incontinence	137	(24.3)	35	(34.0)	0.04	61	(21.0)	7	(18.4)	0.72	76	(27.8)	28	(43.1)	0.02
Grip (kg)															
Right	31.1	(8.9)	29.8	(9.3)	0.21	32.6	(9.6)	34.8	(9.2)	0.20	29.4	(7.7)	26.9	(8.0)	0.02
Left	30.0	(8.7)	28.0	(8.9)	0.04	31.5	(9.4)	32.3	(9.3)	0.62	28.4	(7.6)	25.6	(7.6)	0.01
Social frailty screening index															
Socially robust	238	(42.0)	38	(36.9)	0.55	107	(36.6)	12	(31.6)	0.78	131	(47.6)	26	(40.0)	0.32
Pre-frailty	233	(41.1)	48	(46.6)		134	(45.9)	18	(47.4)		99	(36.0)	30	(46.2)	
Frailty	96	(16.9)	17	(16.5)		51	(17.5)	8	(21.1)		45	(16.4)	9	(13.8)	
Fear of falling	85	(15.0)	38	(36.9)	<0.01	41	(14.0)	8	(21.1)	0.25	44	(16.0)	30	(46.2)	<0.01

^1^ Age, medication, body mass index, albumin and grip are expressed as mean (standard deviation). All others are expressed as number of people (%).

**Table 2 jcm-09-03649-t002:** Relationship between multiple fall history and motor fitness scale in people <65 years old ^1^.

Self-Perceived Motor Fitness	Fall History								
	Model 1 ^2^			Model 2 ^3^			Model 3 ^4^		
	Adjusted OR (95% Cl)		*p* Value	Adjusted OR (95% Cl)		*p* Value	Adjusted OR (95% Cl)		*p* Value
① Able to go up and down stairs	7.01	(0.12–411.16)	0.35	5.84	(0.07–467.19)	0.43	1.51	(0.01–168.70)	0.86
② Able to go up stairs without becoming short of breath	0.79	(0.28–2.26)	0.66	0.94	(0.30–2.96)	0.91	1.07	(0.33–3.55)	0.91
③ Able to jump	2.71	(0.13–54.81)	0.52	1.78	(0.04–70.38)	0.76	0.46	(0.00–48.20)	0.75
④ Able to run	0.00	(0.00– )	1.00	0.00	(0.00– )	1.00	0.00	(0.00– )	1.00
⑤ Able to walk fast to catch up with someone walking	0.34	(0.05–2.20)	0.26	0.28	(0.03–2.34)	0.24	0.19	(0.02–2.41)	0.20
⑥ Able to walk continuously for at last 30 min	0.44	(0.01–20.95)	0.68	0.68	(0.00–92.41)	0.88	1.11	(0.01–215.58)	0.97
⑦ Able to carry a bucket filled with water	0.00	(0.00– )	1.00	0.00	(0.00– )	1.00	0.00	(0.00– )	1.00
⑧ Able to lift a 10 kg bag of rice	0.39	(0.00–30.88)	0.67	0.61	(0.01–60.70)	0.83	1.32	(0.02–107.79)	0.90
⑨ Able to lift a bicycle that has fallen over ^5^									
⑩ Able to open a jar of jam or other food	2.99	(0.53–16.87)	0.21	2.67	(0.38–18.66)	0.32	3.37	(0.44–26.03)	0.24
⑪ Able to touch the floor while standing without bending the knees	1.00	(0.44–2.25)	1.00	0.94	(0.40–2.19)	0.89	0.91	(0.37–2.21)	0.83
⑫ Able to put on socks, pants or a skirt standing up without support	4.18	(1.11–15.80)	0.04	5.96	(1.27–28.00)	0.02	5.60	(1.03–30.55)	0.046
⑬ Able to stand up from a chair without using hands	0.74	(0.04–13.22)	0.83	0.31	(0.01–9.36)	0.50	0.14	(0.00–5.29)	0.29
⑭ Able to sand on tip-toes without holding onto anything	2.63	(0.51–13.65)	0.25	3.20	(0.52–19.80)	0.21	4.69	(0.65–33.78)	0.12

^1^ “Yes” (able to perform the task) is the reference for all items. ^2^ Model 1 is interactively adjusted for the 14 items of the motor fitness scale. ^3^ Model 2 is Model 1 plus an adjustment for sex, education, alcohol consumption, body mass index, types of medication, urinary incontinence and diseases. ^4^ Model 3 is Model 2 plus an adjustment for grip strength, albumin levels, social frailty and fear of falling. ^5^ ⑨ Able to lift a bicycle that has fallen over; All answers are “yes”. The lack of odds ratios is due to the rarity of any of the binary values in the independent variables. OR, odds ratio.

**Table 3 jcm-09-03649-t003:** Relationship between fall history and motor fitness scale in people ≥65 years old ^1^.

Self-Perceived Motor Fitness	Fall History								
	Model 1 ^2^			Model 2 ^3^			Model 3 ^4^		
	Adjusted OR (95% Cl)		*p* Value	Adjusted OR (95% Cl)		*p* Value	Adjusted OR (95% Cl)		*p* Value
① Able to go up and down stairs	0.00	(0.00– )	1.00	0.00	(0.00– )	1.00		(0.00– )	1.00
② Able to go up stairs without becoming short of breath	2.27	(1.20–4.32)	0.01	2.51	(1.21–5.20)	0.01	2.36	(1.08–5.19)	0.03
③ Able to jump	0.48	(0.15–1.62)	0.24	0.44	(0.11–1.71)	0.24	0.45	(0.11–1.91)	0.28
④ Able to run	2.40	(0.77–7.47)	0.13	3.18	(0.88–11.51)	0.08	2.13	(0.54–8.34)	0.28
⑤ Able to walk fast to catch up with someone walking	2.14	(1.01–4.54)	0.048	1.35	(0.59–3.10)	0.48	1.78	(0.73–4.32)	0.20
⑥ Able to walk continuously for at last 30 min	1.51	(0.60–3.80)	0.38	1.58	(0.57–4.35)	0.38	1.28	(0.41–3.99)	0.67
⑦ Able to carry a bucket filled with water	2.17	(0.67–6.98)	0.19	4.17	(1.04–16.72)	0.04	3.95	(0.89–17.57)	0.07
⑧ Able to lift a 10 kg bag of rice	0.43	(0.11–1.68)	0.23	0.20	(0.04–1.00)	0.05	0.15	(0.02–0.91)	0.04
⑨ Able to lift a bicycle that has fallen over	0.92	(0.13–6.39)	0.93	0.57	(0.07–4.72)	0.60	0.47	(0.05–4.85)	0.53
⑩ Able to open a jar of jam or other food	0.92	(0.27–3.11)	0.89	0.62	(0.15–2.59)	0.51	0.50	(0.10–2.48)	0.40
⑪ Able to touch the floor while standing without bending the knees	0.85	(0.45–1.60)	0.61	0.96	(0.47–1.99)	0.92	0.84	(0.39–1.82)	0.66
⑫ Able to put on socks, pants or a skirt standing up without support	2.44	(1.18–5.04)	0.02	3.49	(1.55–7.89)	0.00	3.90	(1.62–9.36)	0.00
⑬ Able to stand up from a chair without using hands	1.38	(0.54–3.48)	0.50	1.36	(0.48–3.85)	0.56	1.34	(0.45–3.96)	0.60
⑭ Able to sand on tip-toes without holding onto anything	0.85	(0.34–2.12)	0.72	0.65	(0.23–1.84)	0.41	0.56	(0.18–1.70)	0.31

^1^ “Yes” (able to perform the task) is the reference for all items. ^2^ Model 1 is interactively adjusted for the 14 items of the motor fitness scale. ^3^ Model 2 is Model 1 plus an adjustment for sex, education, alcohol consumption, body mass index, types of medication, urinary incontinence and diseases. ^4^ Model 3 is Model 2 plus an adjustment for grip strength, albumin levels, social frailty and fear of falling. The lack of odds ratios is due to the rarity of any of the binary values in the independent variables. OR, odds ratio.

**Table 4 jcm-09-03649-t004:** Relationship between multiple fall history and motor fitness scale in people ≥65 years old ^1^.

Self-Perceived Motor Fitness	Multiple Fall History								
	Model 1 ^2^			Model 2 ^3^			Model 3 ^4^		
	Adjusted OR (95% Cl)		*p* Value	Adjusted OR (95% Cl)		*p* Value	Adjusted OR (95% Cl)		*p* Value
① Able to go up and down stairs	0.00	(0.00– )	1.00	0.00	(0.00– )	1.00	0.00	(0.00– )	1.00
② Able to go up stairs without becoming short of breath	2.17	(0.92–5.10)	0.08	2.36	(0.89–6.27)	0.09	1.96	(0.66–5.80)	0.22
③ Able to jump	0.50	(0.10–2.44)	0.39	0.66	(0.12–3.77)	0.64	0.35	(0.05–2.40)	0.29
④ Able to run	1.41	(0.30–6.60)	0.66	1.33	(0.24–7.29)	0.74	1.13	(0.19–6.90)	0.89
⑤ Able to walk fast to catch up with someone walking	1.96	(0.68–5.63)	0.21	1.29	(0.39–4.22)	0.68	1.72	(0.47–6.29)	0.41
⑥ Able to walk continuously for at last 30 min	2.06	(0.64–6.61)	0.23	2.82	(0.77–10.32)	0.12	2.89	(0.65–12.87)	0.16
⑦ Able to carry a bucket filled with water	0.78	(0.18–3.49)	0.75	1.41	(0.27–7.45)	0.68	1.12	(0.17–7.52)	0.91
⑧ Able to lift a 10 kg bag of rice	1.69	(0.35–8.16)	0.51	1.23	(0.20–7.67)	0.82	0.86	(0.09–7.88)	0.90
⑨ Able to lift a bicycle that has fallen over	1.32	(0.17–10.43)	0.79	0.77	(0.07–8.46)	0.83	1.06	(0.07–15.22)	0.97
⑩ Able to open a jar of jam or other food	2.97	(0.73–12.07)	0.13	2.71	(0.52–14.07)	0.24	2.22	(0.34–14.30)	0.40
⑪ Able to touch the floor while standing without bending the knees	1.63	(0.65–4.12)	0.30	1.42	(0.51–3.94)	0.50	1.51	(0.51–4.50)	0.46
⑫ Able to put on socks, pants or a skirt standing up without support	4.27	(1.67–10.94)	0.00	5.36	(1.91–15.06)	0.00	5.41	(1.68–17.43)	0.00
⑬ Able to stand up from a chair without using hands	1.83	(0.55–6.14)	0.33	2.21	(0.55–8.85)	0.26	2.08	(0.48–9.01)	0.33
⑭ Able to sand on tip-toes without holding onto anything	0.73	(0.21–2.51)	0.62	0.47	(0.11–2.06)	0.32	0.34	(0.07–1.72)	0.19

^1^ “Yes” (able to perform the task) is the reference for all items. ^2^ Model 1 is interactively adjusted for the 14 items of the motor fitness scale. ^3^ Model 2 is Model 1 plus an adjustment for sex, education, alcohol consumption, body mass index, types of medication, urinary incontinence and diseases. ^4^ Model 3 is Model 2 plus an adjustment for grip strength, albumin levels, social frailty and fear of falling. The lack of odds ratios is due to the rarity of any of the binary values in the independent variables. OR, odds ratio.

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
