# Peer review of "Relationship between Fall History and Self-Perceived Motor Fitness in Community-Dwelling People: A Cross-Sectional Study"

_jcm, 2020, doi:10.3390/jcm9113649_

Round 1

Reviewer 1 Report

Dear authors,

thank you for adaptation of this socially relevant problem.

You found out for yourself some limitations of the study. In my opinion the estimation of motor fitness by a self-report (questionnare) is particularly uncertainly. I would strongly recommend to use physical activity tests (e.g. Timed Up and Go or Short Physical Performance Battery; s. Liu-Ambrose et al., 2019).

Furthermore, I could not find in your analysis some essential factors as fall risks, which can evoke a fall: visual impairment, medications, muscle weakness, static and dynamic balance.

In addition yous should discuss the influence of physical activity to fall history. We can assume, that a correlation exists between motor fitness and physical activities or sports activities of older people. This would be helpful to find future interventions.

Author Response

Thank you for your constructive comments and useful suggestions, which have improved our paper. As indicated in the responses below, we have revised the paper according to your comments and suggestions.

You found out for yourself some limitations of the study. In my opinion the estimation of motor fitness by a self-report (questionnare) is particularly uncertainly. I would strongly recommend to use physical activity tests (e.g. Timed Up and Go or Short Physical Performance Battery; s. Liu-Ambrose et al., 2019).

Reply:

Thank you for your advice.

We have included the need for physical activity tests such as Timed Up and Go or Short Physical Performance Battery in the limitations. (p.11, line 308-315)

Furthermore, I could not find in your analysis some essential factors as fall risks, which can evoke a fall: visual impairment, medications, muscle weakness, static and dynamic balance.

Reply:

Thank you for your feedback.

In this study, the type of medication used and grip strength were adjusted in the analysis; however, indicators for lower limb muscle strength, visual impairment, static and dynamic balance were not used. We have added to this in the limitation. (p.11, line 308-315)

This study was conducted using a self-report. Significantly, some of the motor fitness tasks reflected in daily life activities required clarification.

In addition you should discuss the influence of physical activity to fall history. We can assume, that a correlation exists between motor fitness and physical activities or sports activities of older people. This would be helpful to find future interventions.

Reply:

Thank you for your advice.

We have included information regarding the relationship between motor fitness and physical activities in the discussion. (p.10, line 266-268)

Reviewer 2 Report

Dear authors

Thank you for conducting a study on the important matter of fall history and possible easy to detect fall risk factors. I really like the idea of including younger subjects (40 to 65) into this study. This could allow for possible early detection of fall risk. I have some suggestions that you can find in the attached file.

Kind regards

Author Response

Thank you for your constructive comments and useful suggestions, which have improved our paper. As indicated in the responses below, we have revised the paper according to your comments and suggestions.

Major points:

  1. Your abstract and some sections of the discussion attempt to bridge the gap from an association between parameters in a cross-sectional setting to an effective intervention idea (correctable-fall-related factors). I would advise against exploring this line of thought, as associations in present variables and associations in changes in variables should be treated differently as they are not the same. When one variable is associated with another this does not mean that a change over time in said variable is also associated with a change in the other variable. Hence, the general idea to identify associations with fall risk to some measurable variable in order to hopefully target that variable in interventions and hoping to also influence fall risk is unwarranted in my opinion. I would prefer that those sections are reconsidered and a focus on the observed associations and the possible predictive quality of the MFS to detect people at a risk of falling.

Reply:

Thank you for your valuable advice.

We considered that focus on the observed associations and the possible predictive quality of the MFS to detect people who are at risk of falling rather than mentioning the possibility of intervention. (abstract, introduction, discussion, and conclusion)

  1. In your results, you never analysed the association between overall MFS score and fall history. In your method section you provide some cut offs for the MFS in relation to requiring nursing care. As your results suggest, only two (three if we would only look at model 1) items of the MFS appear to be relevant for history of falling. Also, in your discussion you state that motor fitness is “strongly” associated with fall history, yet this is not backed up by your data. Only very few of your MFS items are associated with fall history. I would strongly suggest sticking to the calculated results and be more cautious with broadening those results to the whole MFS.

Reply:

Thank you for pointing out this issue.

“Only few of the MFS items are associated with a fall history.” We avoided broadening those results to the entire MFS. (discussion and conclusion)

  1. If the MFS is largely unassociated with falls history, and your stated relation with physical performance assessments is excellent, one would expect there to also be no association between physical performance and fall history. I would like to see more caution while comparing MFS and physical performance assessment from your source 15. The association within this study is significant, but with r=0.59 not very strong. Especially, considering the investigated population was different (only >65 years). As physical performance differences between subjects tend to increase with older age, the investigated sample was probably more heterogenous than your sample of 40 to 65-year olds. For the sample of 40 to 65-year olds, the association between MFS and physical performance might be weaker and thus I would be more cautious in proclaiming “equality” of measurements unless validity data for this age range could be provided.

Reply:

Thank you for your advice.

We have modified the text as follows: The predictive power of physical function assessed by the MFS is related to the assessment of physical performance in community-dwelling older people (p.3, line 101-102)

We have deleted “equality” in the methods. Furthermore, deleted “However, MFS has been shown to reflect physical function [15]” in the limitations.

Minor points:

  1. In Tables 2, 3 and 4 there is an Odds Ratio of Zero for some items (Table 2: 4, 7; Table 3: 1; Table 4: 1) but no explanation for this lack of Odds Ratio is provided.

Reply:

The lack of odds ratios is due to the rarity of any of the binary values in the independent variables.We have stated this in the notes of Tables 2 and 3.

  1. I personally, when reading the term “motor fitness scale”, have a very different connotation expecting some physical performance test battery. Maybe, consider using something like “self-perceived motor fitness scale” or similar, to indicate the subjective nature of the method.

Reply:

Thank you for your comment.

We have replaced the term “motor fitness scale” with ”self-perceived motor fitness scale” in the entire manuscript including the title. We defined MFS as self-perceived motor fitness scale at the first mention, and thereafter we used the “MFS”.

  1. You state that motor fitness can also account for responses towards fall anxiety, yet no analysis in this regard was provided. Yes, looking at the descriptive data this might be warranted, but a model exploring this association is lacking.

Reply:

We stated that motor fitness can also account for responses toward fall anxiety. Because no analysis was provided in this regard, we have deleted the following text from the manuscript: “Furthermore, fall history is strongly associated with anxiety towards falls, which limits daily life [8]. Motor fitness can also account for responses toward fall anxiety.”

  1. You state that changing clothes is an observable event associated with the history of falls and it can be observed by family members. In general, this study gives an indication of what queues to look for in adults to assess if they might be at a risk of falls or have a fall history. I was wondering whether changing clothes is routinely done with family members present in Japan, or if this would only apply to a nursing home or care setting.

Reply:

Thank you for your comment.

In accordance with your suggestion, changing clothes is not routinely carried out in the presence of family members.

We have deleted “by family members.” (p.10, line 265-266)

  1. I have a hard time understanding the sentence starting in line 64. Please consider rewording.

Reply:

Thank you for your feedback. We have modified the text as follows: If a relationship between fall history and some motor fitness can be identified, then older people may be able to easily identify fall-related factors in their lives (p.2 line 66-67)

  1. Please indicate how grip strength was measured in regard to body position of the subject and angle of the elbow.

Reply:

Grip strength was measured alternately in the left and right hands in the standing position with the elbow extended. (p.3, line 112-114)

  1. For table 1, when it extends beyond 1 page, I would add the heading (row 1 and 3 – age group, fall history) on the second page again for better readability. Roughly the same applies to all other tables.

Reply:

As advised, we have added the heading.

Round 2

Reviewer 1 Report

no comments

Author Response

Comments and Suggestions for Authors: no comments

Thank you for your careful peer review.

Reviewer 2 Report

Thank you for considering my inputs and substantially improving the manuscript. I feel most points and the Overall trajectory of the paper has been adequately changed.

I have some minor points left:

  • In line 109, you mention an "MTF". It is unclear to me, what this is.
  • In line 229, I would remove the subclause "which is associated with fall history," as this constitutes a broadening of the results to the whole MFS instead of only a few subitems, as discussed appropriately later. If Motor fitness is in this sentence should understood in a different context (physical functioning) please indicate so.

Kind regards

Author Response

Thank you for your constructive comments and useful suggestions, which have improved our paper. As indicated in the responses below, we have revised the paper according to your comments and suggestions.

Thank you for considering my inputs and substantially improving the manuscript. I feel most points and the Overall trajectory of the paper has been adequately changed.

I have some minor points left:

In line 109, you mention an "MTF". It is unclear to me, what this is.

Reply:

Thank you for your comment.

This information was not necessary and has been removed.

In line 229, I would remove the subclause "which is associated with fall history," as this constitutes a broadening of the results to the whole MFS instead of only a few subitems, as discussed appropriately later. If Motor fitness is in this sentence should understood in a different context (physical functioning) please indicate so.

Reply:

Thank you for your valuable advice.

In accordance with your suggestion, we have deleted this subclause.

This manuscript is a resubmission of an earlier submission. The following is a list of the peer review reports and author responses from that submission.

Round 1

Reviewer 1 Report

I would like to thank the authors for a very interesting manuscript. I have some serious doubts that need clarification (see comments below). 

A general comment is to review the language so that the English runs smoothly throughout. Please also avoid concepts such as Elderly. This is known to be stigmatizing and could preferably be changed to Older people. 

Also, the use of the word Correctable is confusing. Does it mean that the risk factor can be changed by means of an intervention?

Please also provide page number to all references to quotes. 

Specific comments:

Introduction:

There is no rationale presented to the division of the participants in two groups for the statistical analysis. Please provide. 

No description of what constitutes daily activities is presented which makes the reading and evaluation of the manuscript difficult. a 

Methods:

Please switch places between 2.2 and 2.3 so that Basic attributes comes first

Was the Japanese version used for all measurement instruments? Please state. 

In several of the descriptions of data collection instruments, results of previous studies (eg. relationships between falls history and fear of falling) is presented. Please move such information to the rationale/introduction.

In the aim of the study it is stated that daily activities is a key variable, however, there is no explanation or instrument which the authors define or describe  measures of daily activities. From the analyses one can conclude that it maybe is measured by the MFS, but this is not clear. In this context it is not clear what the other instruments measure and how this is related to daily activities. This needs to be clarified. Maybe the use of additional headings would help? Such as Daily activities (and then the instruments measuring this variable could be stated under subheadings) and Falls history?

Results: 

Again, without a clear description of what constitutes daily activities (no rationale is provided) it is difficult to evaluate the results.

Table 1 can be moved to the Methods section, given that it comprises only Basic attributes (called participant characteristics in the table). I can see the intention of the authors here but it is really clear to the reader. 

Discussion: 

It is difficult to comment on the discussion since the term daily activities are used here again while there is no description of what it means and how it is measured. A few comments, however: 

I question why the discussion is built up item by item from the MFS scale? This is probably not an efficient way to discuss the findings. 

The discussion comprise very little discussion of the findings in the light of research literature. Instead, the repetition of results takes up too much space. 

I also question the statement under Limitations that the findings cannot be generalized due to the fact that the population was healthy. I can see where the authors are heading but a more thorough explanation is required. 

Reviewer 2 Report

The article submitted by Yokoi et al. aims to describe the relationship between fall history and daily activities among community-dwelling middle- and older-aged adults. In general, the article addresses an important question. However, this manuscript has important limitations especially in relation to the methods used and the analysis performed in the study.

The main issue is about the data collection, specifically about the date of measurement of the fall history. History of falls and questionnaires are measured at the same time. This is a major limitation that skews the study data and conclusions. The results found in the variables from the questionnaires and handgrip may be conditioned since participants have fallen previous to the measurements. Therefore, these questions or test are not predictors of falls, but maybe a consequence of the falls produced. For example, a negative answer to the question “Able to put on socks, pants, or a skirt while standing without support” may be determined because the individual has fallen several times and therefore is not able to carry out those tasks or is capable but with more difficulties. Maybe, if the question had been asked before the fall, then the individual would have not answered negatively. In my experience, when I analyze the risk of hospitalization, I do so by measuring the variables that I believe can be predictors beforehand. Then, a follow-up of these hospitalizations is carried out. The same thing would happen with the risk of falls. Conclusions like the one made here "These daily activities can be easily noticed by an individual or people close to them and may help them become aware of the risk of falling earlier" are inappropriate because as I explained previously, that feeling is not a predictor of the fall, but a consequence.

Another major aspect is that all measurements except the handgrip are self-reported. Furthermore, the analyses are not corrected for physical activity, lower limb strength or cognition, variables that could more fully explain the models, and subtract the strength of the association from the self-reported variables.